# *Diplosphaera elongata* sp. nova: Morphology and Phenotypic Plasticity of This New Microalga Isolated from Lichen Thalli

Salvador Chiva [1,2,*], César Daniel Bordenave [2], Ayelén Gázquez [2] and Eva Barreno [2]

1 Department of Life Sciences, University of Trieste, Via L. Giorgieri 10, 34127 Trieste, Italy
2 Fac. CC. Biológicas, Instituto Cavanilles de Biodiversidad y Biología Evolutiva, ICBIBE, Universitat de València, C/Dr. Moliner, 50, 46100 Burjassot, Spain
* Correspondence: salvador.chiva@uv.es

**Abstract:** Lichen phycobiomes have recently emerged as a source of biodiversity and new species of microalgae. Although in the genus *Diplosphaera* free-living microalgae are common, numerous strains belonging to this genus have frequently been recognized or isolated from lichen thalli. In this study, a comprehensive analysis of the strain *Diplosphaera* sp. ASUV135, isolated from a lichen thallus, has been carried out using an integrative taxonomic approach. The SSU and nuclear-encoding ITS rDNA, as well as the chloroplast *rbc*L gene, were sequenced and analyzed to ascertain its taxonomic position and phylogenetic relationships within the genus *Diplosphaera*. This strain was also analyzed by light, confocal and transmission microscopy for morphological and ultrastructural characterization. The phenotypic plasticity in this strain was also confirmed by changes in its morphology under different growth conditions, as well as those of modulated Chlorophyll *a* fluorescence emissions, to understand its photosynthetic functioning. Our results pointed out that this strain represents a new taxon within the genus *Diplosphaera* (*Prasiola* group), described here as *Diplosphaera elongate* sp. nova. This study also provides tools for future research on organisms with high phenotypic plasticity by using molecular, morphological, ultrastructural and physiological approaches.

**Keywords:** description; modulated fluorometry; phycobiome; phylogeny; *Stichococcus*-like organism; ultrastructures

## 1. Introduction

The *Prasiola* clade is included as a Trebouxiophyceae lineage, showing great morphological diversity with different cellular organization levels within its members [1]. Recently, phylogenetic relationships were clarified, and several new genera were described in this clade [2–5]. At last, the green puzzle of *Stichococcus*-like organisms [5], composed mainly of microalgae indistinguishable under microscopy, has been solved. This earlier study clarified the phylogenetic position of *Desmococcus*, *Diplosphaera*, *Pseudostichococcus* and *Stichococcus* and proposed four new genera, *Deuterostichococcus*, *Protostichococcus*, *Tetratostichococcus* and *Tritostichococcus*, to be included in the *Prasiola* clade [5]. The microalgae of this group of genera related to *Stichococcus* are widely distributed in freshwater, marine and terrestrial ecosystems [6–8], as well as in harsh polar environments [9–11]. Morphologically, they are characterized by cylindrical or short-cylindrical cells containing a plate-like chloroplast, mostly without a pyrenoid [5,12].

The taxa of the *Diplosphaera* genus have also been clarified in Pröschold and Darienko [5]. Up until that publication, three species were recognized for this genus in the Algae-Base [13]: *Diplosphaera chodatii* Bialosuknia (genus type), *Diplosphaera epiphytica* Darienko and Pröschold and *Diplosphaera mucosa* Broady. After this work [5], it was shown that, using integrative approaches combining molecular phylogenetic and morphological analysis, all the studied strains belonged to the same species. *Diplosphaera epiphytica* was synonymized with *D. chodatii*, and *Diplosphaera mucosa* became a variety of *D. chodatii*, namely *D. chodatii* var. *mucosa* (Broady) Pröschold and Darienko.

*Diplosphaera* strains exhibit a wide variety of morphologies including micareoid, globose (ovular/spherical) and cylindrical shapes [14–17]. These microalgae, when showing a globose form, typically clump two to four cells together [5,18,19]. The phenotypic plasticity characteristic of many *Stichococcus*-like organisms has also been studied in strains of the genus *Diplosphaera*. Bialosuknia [18] subjected *D. chodatii* strains to different culture media so as to observe different nutritional behaviors. This author described this microalga as globose, 2 μm in diameter, but its size can vary depending on the medium in which it grows. Different colony sizes and colorations were also observed depending on the medium. Pröschold and Darienko [5] compared the cellular shapes of six *D. chodatii* strains (SAG 11.88, SAG 2049, SAG 49.86, SAG 48.86, SAG 2.82 and SAG 9.82) in different culture media and temperature conditions. They concluded that at a low temperature (8 °C) all the tested strains of *D. chodatii* showed elongated forms and that phenotypic plasticity is important for taxonomic circumscription in microalgae.

The microalgae of the genus *Diplosphaera* have been correlated with lichen symbioses from the onset. Bialosuknia [18] described the genus with strains isolated from the lichen *Lecanora tartarea* (now *Ochrolechia tartarea* (L.) A. Massal.). Subsequently, several studies have also detected microalgae of the genus *Diplosphaera* growing on lichens; see [20] and references therein. The above-cited study notes that the main fungal partners of *Diplosphaera* are members of the Verrucariaceae family [20]. The semi-aquatic lichen *Dermatocarpon* in North America and Europe is a widely studied example of a lichen-forming fungus associated with microalgae of the genus *Diplosphaera* [15,21]. The presence of *Diplosphaera* spp. has also been reported as a photosymbiont partner in the lichen genera *Staurothele* and *Endocarpon* [15,21,22]. Although it is often not specified whether it is the primary microalga, it may often be that *Diplosphaera* spp. could be part of the lichen phycobiome, as already reported for *Stichococcus*-like microalgae isolated from lichens [23]. Symbiosis is a lifestyle common to many other microalgae of the *Prasiola* clade, especially in the case of the lichen family Verrucariaceae [22]. Another example of the Verrucariaceae/*Prasiola* clade is the association between *Mastodia* spp. (mycobiont) and *Prasiola* spp. (photobiont), where the mycobiont grows within the independently formed algal thallus [24,25]. Moreover, lichenized *Prasiola* clade algae have also been recorded in the lichen genera *Polyblastia*, *Agonimia*, *Normandina*, *Catapyrenium*, *Placopyrenium*, *Placidiopsis* and *Neocatapirenium* [22]. Recently, in the lichen family Agyriaceae, the microalga *Deuterostichococcus allas* was identified as a symbiont of the lichen *Placopsis* in Antarctic populations [26].

The aim of our study was to develop a comprehensive analysis of the microalgal isolates of a *Buellia zoharyi* lichen phycobiome using an integrative approach. We sequenced the SSU and nuclear-encoding ITS rDNA—the chloroplast *rbc*L gene of the selected strains—and characterized their ultrastructures and morphologies with light, confocal and transmission microscopies, as well as their photosystem II performance through chlorophyll *a* fluorescence studies to ascertain their taxonomic status and better understand their physiological behavior. Our results indicate that this isolate represents a new taxon within the genus *Diplosphaera* (*Prasiola* group) and it is described here as *Diplosphaera elongata* sp. nova. This study also provides tools to study organisms with high phenotypic plasticity using molecular, morphological, ultrastructural and physiological approaches and, thus, assess whether phenotypic plasticity and photosynthetic measurements can be considered important characteristics for the taxonomic circumscription of organisms with similar morphologies, such as *Stichococcus*-like organisms.

## 2. Materials and Methods

### 2.1. Strain Origin and Culture Conditions

The investigated strain, ASUV135, from the Symbiotic Algal collection at the Universitat de València (ASUV, Spain), was isolated using the protocol described in Chiva et al. [27] from a thallus of the lichen *Buellia zoharyi* [23]. In the latter study [23], the isolated strain ASUV135 was identified by ITS rDNA phylogeny as a microalga of the genus *Diplosphaera*, named *Diplosphaera* sp. ASUV135.

Another strain, *Diplosphaera chodatii* CCAP 416/1, isolated from a water bloom from a small bog in Basilea, Bot. Gard. Univ. de Basilea (Culture Collection of Algae and Protozoa (CCAP, Scotland)), was used in this study for comparison with *Diplosphaera* sp. ASUV135. The former was referenced as the "Equivalent Strain" of *D. chodatii* SAG 49.86 (epitypic strain of *Diplosphaera chodatii* Bialosuknia emend. Vischer [5]) in the CCAP catalog (https://www.ccap.ac.uk/catalogue, accessed on 22 January 2023). Both strains were maintained under a 50 µmol/m$^{-2}$ s$^{-1}$ photosynthetic photon flux density (PPFD) with a light:dark cycle of 16:8 hrs. at 18 °C in liquid Bold's Basal Medium (BBM) [28].

To test the phenotypic plasticity, *Diplosphaera* sp. ASUV135 was grown on a standard medium (3-fold nitrogen-BBM + vitamins = 3N-BBM + V, [29]), a poor medium (BBM, [28]), a rich medium (3-fold nitrogen-BBM + glucose and caseine = 3N-BBM + GC, [30]) and a medium supplemented with lichen extract (3-fold Nitrogen-BBM + lichen extract = 3N BBM + LE, described here; see Appendix A). Three replicates per medium condition were grown in both liquid and solid mediums (with agar) at 8 °C and at 18 °C; a total of sixteen treatments were tested (Table 1). To compare *D. chodatii* CCAP 416/1, 3N-BBM + V, BBM and 3N-BBM + GC media at 8 °C were used in solid and liquid media. Luminosity and light/darkness parameters in the growth chambers were the same as described above.

**Table 1.** Treatments used to test phenotypic plasticity. All the treatments were subjected to 50 µmol/m$^{-2}$ s$^{-1}$ PPFD with a light:dark cycle of 16:8 h. for 21 days. Abbreviations: 3N (3-fold nitrogen), BBM (Bold Basal Medium), V (vitamins), GC (glucose and caseine) and LE (lichen extract). See Appendix A.

| Treatment | Standard Condition | Poor Condition | Rich Condition | Lichen Condition |
|---|---|---|---|---|
| Solid 8 °C | 3N-BBM + V | BBM | 3N-BBM + GC | 3N-BBM + LE |
| Solid 18 °C | 3N-BBM + V | BBM | 3N-BBM + GC | 3N-BBM + LE |
| Liquid 8 °C | 3N-BBM + V | BBM | 3N-BBM + GC | 3N-BBM + LE |
| Liquid 18 °C | 3N-BBM + V | BBM | 3N-BBM + GC | 3N-BBM + LE |

*2.2. Microscopic Analyses*

The morphological observations were carried out on twenty-one-day-old cultures. For the differential interference contrast (DIC) microscopic investigations of *Diplosphaera* sp. ASUV135, a Nikon Eclipse E-800 microscope (Nikon, Tokyo, Japan) was used with a Nikon DS-Ri1 camera (Nikon, Tokyo, Japan). For each treatment, four independent samples were observed. For each sample, 4 nonoverlapping fields were imaged. Non-supervised analysis of the images was performed with the Fiji distribution of ImageJ [31], including the following steps: despeckle, Median Filter (radius = 2), Gaussian Blur Filter (sigma = 2), Morphological Filter (operation = Gradient element = Disk radius = 5), Auto Threshold with Triangle Method, Binary Watershed. The Particles8 plugin was exploited to exclude small debris and clumps of cells (based on the cell size previously observed). Area, circularity, Feret diameter, aspect ratio, roundness and solidity were recorded for each cell, calculated as described in ImageJ Documentation (https://imagejdocu.list.lu/, accessed on 22 January 2023). Aspect ratio (AR) was selected as the best-suited variable to assess cylindrical cell morphology. AR was calculated as the Feret diameter (longest internal diameter)/longest internal diameter perpendicular to the Feret diameter. The minimum AR is around one, which is when the cells are spherical (equal in width and length), and when the cells are more elongated (different in length and width), the value is greater than one. For DIC microscopy, all the medium conditions described in Table 1 were used. For morphological comparison of *D. chodatii* CCAP 416/1 with *Diplosphaera* sp. ASUV135, 3N-BBM + V, BBM and 3N-BBM + GC media at 8 °C in solid and liquid mediums were used.

For confocal laser scanning microscopy (CLSM), the *Diplosphaera* sp. ASUV135 cultures were processed as described in Bordenave et al. [32]. Briefly, the whole algal colony was scraped off from the solid medium with a sterilized loop and resuspended in 30 µL of

sterilized liquid medium. In total, 15 μL of each suspension was placed on plates with a thin layer of 1% agar in sterile water, air-dried and placed upside down over a 35 mm imaging dish suitable for inverted microscopy. For the liquid medium experiments, 15 μL of each liquid culture was directly used as described above. An Olympus FLUOVIEW FV1000 laser scanning confocal microscope (Olympus, Tokyo, Japan) was used with a 488 nm excitation laser. Fluorescence emitted from 650 to 750 nm was collected to observe chlorophyll autofluorescence, thus recovering the chloroplast layers. A series of images was captured with a separation of 0.4 μm. The image stack was preprocessed to remove noise and then analyzed using the z-projection tool and volume viewer with the Fiji distribution of ImageJ [31]. For CLSM conditions, BBM and 3N-BBM + GC media at 8 °C and 18 °C (solid and liquid) were used.

For transmission electron microscopy (TEM) analyses, each sample of *Diplosphaera* sp. ASUV135 was fixed and dehydrated as described in Bordenave et al. [32]. In brief, samples were fixed in 2% Karnovsky fixative for 12 h at 4 °C, washed three times for 15 min with 0.01 M PBS (pH 7.4) and postfixed with 2% $OsO_4$ in 0.01 M PBS (pH 7.4) for 2 h at room temperature. After washing in 0.01 M PBS, pH 7.4, the samples were dehydrated at room temperature in a graded series of ethanol starting at 50% and increasing to 70%, 95% and 100% for no less than 20–30 min at each step. The fixed and dehydrated samples were embedded in Spurr's resin according to the manufacturer's instructions (http://www.emsdiasum.com/microscopy/technical/datasheet/14300.aspx, accessed on 22 January 2023). Ultrathin sections, 80 nm thick, were cut with a diamond knife (Diatome ultra 458), mounted on 100 mesh copper grids and then stained with 10% uranyl acetate and 0.1% lead citrate using the "Synaptek Grid-Stick Kit". Sections were observed at 80 kV under a JEOL JEM-1010 microscope (Jeol, Peabody, MA, USA). Images were obtained using an Olympus MegaView III camera and processed with the Fiji distribution of ImageJ [31]. For TEM analysis, the solid BBM medium condition at 18 °C was used.

### 2.3. DNA Extraction

To perform microalgae identification, the total genomic DNA of the unialgal liquid cultures was isolated and purified using the DNeasy TM Plant Mini Kit (Qiagen, Hilden, Germany) following the manufacturer's instructions. PCR amplification was performed in 50 μL using the EmeraldAmp GT PCR Master Mix (Takara), which required the addition of the template DNA, specific primers and water. Three algal loci were amplified; the nuclear SSU rDNA was amplified using the primers 18F and 18R [33] and the nuclear ITS rDNA, including ITS1, 5.8S rDNA and ITS2, using the primer pair nr-SSU-1780 [34] and ITS 4T [35]; and the plastidial *rbc*L was amplified using the primers *rbc*L151f and *rbc*L986R [36]. PCR reactions for SSU and ITS were performed as described in Chiva et al. [27] and for *rbc*L, as described in Nelsen et al. [36]. Amplifications were carried out on a 96-well SensoQuest Labcycler (Progen Scientific). The PCR products were visualized on 2% agarose gels and purified using the Gel Band Purification Kit (GE Healthcare Life Science, New York, NY, USA). The amplified PCR products were sequenced with an ABI 3100 Genetic Analyzer using the ABI BigDyeTerminator Cycle Sequencing Ready Reaction Kit (Applied Biosystems, Waltham, MA, USA).

### 2.4. Phylogenetic Analysis

All the sequences obtained from the new strains were compared with each other and all the known sequences in the GenBank by using the BLAST algorithm for precise identification [37].

Since the *rbc*L dataset could not be concatenated with SSU and ITS, due to having few sequences in common and being well-named in the GenBank (Supplementary Table S1), two phylogenetic analyses of *Stichococcus*-like organisms were performed, one of them with the concatenation of SSU and ITS sequences from the GenBank and those obtained from the strains analyzed, and the other with the *rbc*L sequences from the GenBank and the newly obtained ones. Each dataset was aligned using MAFFT v.7.402 software [38].

Ambiguous regions of both alignments were systematically identified and removed using the program Gblocks v.0.91b [39] and then modified manually by using MEGA7 [40]. Four sequences of *Pseudostichococcus* were selected as the outgroup in the SSU + ITS alignment and *Stichococcus bacillaris* (SAG 379–1b) in the *rbc*L alignment.

The most appropriate nucleotide substitution model for SSU, ITS and *rbc*L (GTR + G, GTR + I + G and GTR + I + G, respectively) was chosen under the corrected Akaike information criterion using JModelTest v.2.1.6 [41]. The phylogenetic trees of both loci were inferred using Bayesian inference (BI) and maximum likelihood (ML) approaches. ML analysis was implemented in RAxML v.8.2.12 [42] using the GTRGAMMA substitution model with 1000 bootstrap pseudoreplicates [43]. BI was carried out in MRBAYES v.3.2.7a [44]. There were two parallel Markov Chain Monte Carlo (B/MCMC) algorithms running with 6 chains simultaneously, each initiated with a random tree, for 10 million generations, and the trees were sampled every 100 generations for a total sample of 200,000 trees. The phylogenetic tree was visualized in FIGTREE v.1.4.3 [45]. All analyses were run at the CIPRES Science Gateway v.3.3 webportal [46].

### 2.5. Analysis of the ITS2 Secondary Structure

The coding regions, required for the basal stems in ITS2 secondary structures, were delimited by a comparison between our sequences and the sequence of the epitypic strain of *Diplosphaera chodatii* (SAG 49.86) [5]. The ITS2 sequences were annotated using the ITS2 database [47]. The minimum energy secondary structure model of ITS2 was computed with RNAstructure v.5.3 [48], and the resulting image was processed using Inkscape (www.inkscape.com, accessed on 22 January 2023).

### 2.6. Chlorophyll-Fluorescence Measurements

Chlorophyll *a* fluorescence emissions were measured using a pulse amplitude-modulated fluorometer (PAM-2000, H. Walz, Effeltrich, Germany). The maximum quantum efficiency of PSII ($F_v/F_m$) was determined using the saturation pulse technique [49]. For this, after a period of dark acclimation (30 min) to allow for the complete oxidation of the PSII reaction centers (all centers are open), the minimum fluorescence yield ($F_0$) was measured. The maximum fluorescence ($F_m$) was determined by applying a saturating pulse of 800 ms (intensity = 10) when all primary electron acceptors $Q_A$ were reduced by the light (all centers are closed). For the slow induction kinetics, samples were illuminated by an actinic light of 500 μmol photons $m^{-2}$ $s^{-1}$ for 330 s, starting 40 s after the saturating pulse. Transient fluorescence yield was followed by the application of a saturation pulse every 20 s. A 3 s pulse of far-red light (intensity = 6) with peak emission at 735 nm was used for pre-illumination to pre-oxidize the PSII acceptor pool before the assessment of the minimal fluorescence yield of the pre-illuminated samples, and preceding the recording of induction kinetics. Light energy absorbed by PSII has different fates, which competitive processes contribute [50,51]. It can be used to drive photochemistry (photochemical quenching), and then, electrons would be transferred from chlorophyll reaction center P680 to the primary quinone acceptor (QA). Alternatively, light energy can be lost due to fluorescence or heat, constituting the non-photochemical quenching. The effective quantum yield ($\varphi_{PSII}$), coefficient of photochemical quenching (qP) and non-photochemical quenching parameter (NPQ) were determined according to Schreiber [49]. For chlorophyll *a* fluorescence emissions, all the solid medium conditions described in Table 1 were used.

### 2.7. Statistical Analyses

The data of cell morphology and chlorophyll-fluorescence were analyzed with two-way analysis of variance (ANOVA) followed by post hoc comparisons using Tukey's HSD test. The value 0.05 was considered the significance level. Calculations were performed using R v.4.0.5 [52].

## 3. Results

*3.1. Taxonomic Assessment*

- Family: Trebouxiaceae Friedl
- Genus: Diplosphaera Bialosuknia
- *Diplosphaera elongata* Chiva and Barreno sp. nova.

Etymology: (e.lon'ga.ta.) L. fem. part. adj. elongata; elongated, stretched out, pertaining to the elongated cell shape.

Description: Rod-shaped (cylindrical) cells, highly variable in size. They range in length from 1.77 to 33.92 μm and in width from 1.67 to 4.27 μm. In rich media, they have abundant oil vesicles and may also have pear-shaped cells. The chloroplast is usually situated in a longitudinal position close to the cell wall with irregularly arranged thylakoid membranes; no pyrenoid was observed. In the long cells, two or more chloroplasts may be present. The cell wall shows two layers, the inner one being thinner than the outer one. Although they break easily, they can form long filaments (2–10 cells) as tangled threads. Reproduction by fragmentation of filaments due to vegetative cell division.

Diagnosis: Differs from the *D. chodatii* (epitype, SAG 49.86) in the cell form. Individual cells from *D. chodatii* are rather oval, whereas the cells of *D. elongata* are rod-shaped. Moreover, the species differs from the *D. chodatii* (epitype, SAG 49.86) in the *rbc*L chloroplast gene and the 18S-ITS1–5.8S-ITS2 rDNA nuclear genes (including the ITS2 secondary structure). They can also be distinguished by their phenotypic behavior under different growth conditions. *D. chodatii* cells, at 8 °C, have a slightly oval form. In contrast, *D. elongata* elongates its cells greatly in sugar-rich media, but it is not significantly affected by temperature.

Type locality: Isolated from the lichen *Buellia zoharyi* Galun, collected on a volcanic substrate in Igueste de San Andrés (Tenerife, Canary Islands), specifically, on intermediate castings and mafic phonolites; 28° 31′ 44″ N 16° 08′ 49″ W (leg. Arnoldo Santos, 2014).

Holotype (designated here): Cryopreserved cells of strain *Diplosphaera* sp. ASUV135 at the Symbiotic Algal collection of the Universitat de València (ASUV), item TYPE–ASUV 135.

Reference strains: ASUV 135 and ACOI 3417, deposited at the Coimbra Collection of Algae (ACOI), University of Coimbra.

Iconotype (designated here to support this holotype): Figures 1 and 2 in this study.

Ecology and distribution: At the time of writing, it has been detected in the following lichen species:

*Agonimia tristicula*, on mosses on carbonaceous rock, south of Gaberl Pass, Stubalpe, Styria, Austria (GenBank code JN573847 [22]).

*Endocarpon pusillum*, on soil, Plaine de la Crau, Bouches-du-Rhônes, France (GenBank code JN573833 [22]).

*Staurothele frustulenta*, on siliceous rock, dyke along Ijsselmeer, Netherlands (GenBank codes JN573824 and JN573890 [22]).

*Ramalina farinacea*, in a pine forest, El Toro, Castellón, Spain (GenBank codes OK636220 and OK636221 [23]).

*Parmotrema pseudotinctorum*, on ancient basaltic rocks, San Sebastián, La Gomera, Spain (GenBank codes OK636234 and OK636235 [23]).

Moreover, this microalga has also been detected as free-living in plantations of *Pinus radiata* located inside the volcanic plateau of North Island in New Zealand (38° 31′ S 176° 34′ E; GenBank code KX221524 [53]) and in a soil sample (GenBank code MF481512, direct submission).

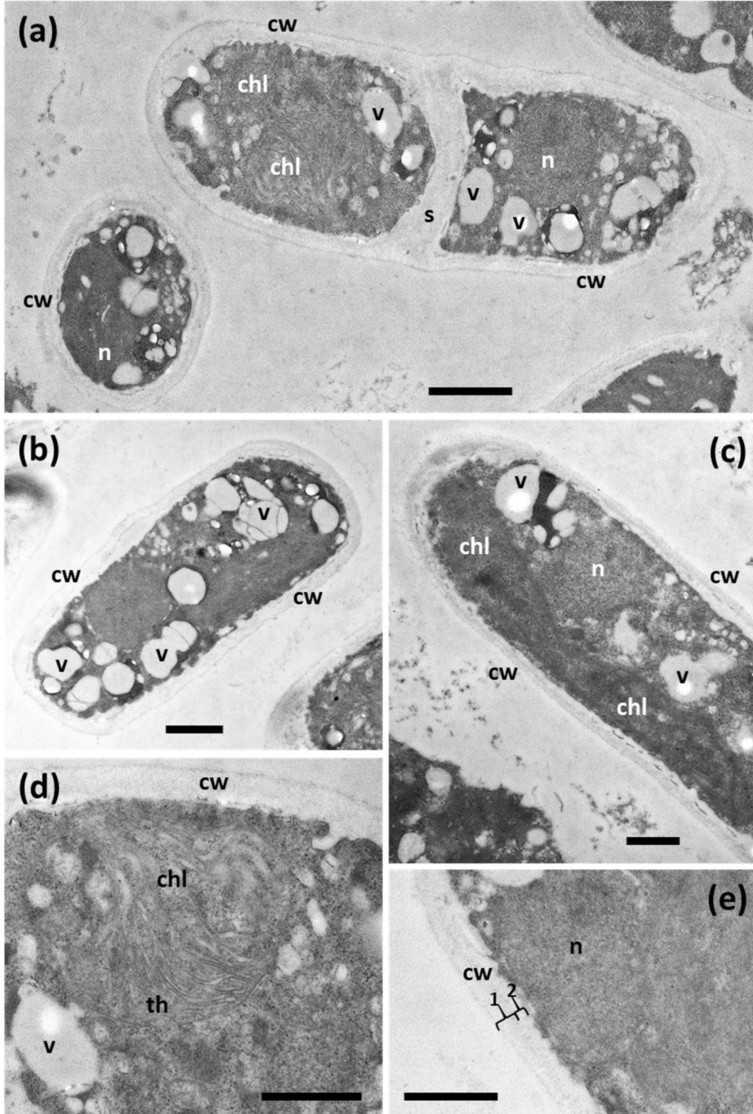

**Figure 1.** *Diplosphaera elongata* sp. nov. ultrastructural features found using TEM growing in solid BBM at 18 °C: (**a**) *D. elongata* cells—on the right, a cell cross-section, and on the left, a longitudinal section of a divided cell; (**b**) single rod-shaped cell; (**c**) chloroplast situated in a longitudinal position close to the cell wall; (**d**) detail of the chloroplast and thylakoids; (**e**) detail of the cell wall showing two layers (1,2). Abbreviations: chl (chloroplast), cw (cell wall), n (nucleous), s (septum), th (thylakoid), v (vesicles). Scalebars: (**a**) = 1 μm, (**b**) = 800 nm, (**c**–**e**) = 600 nm. Reference strain: ASUV 135.

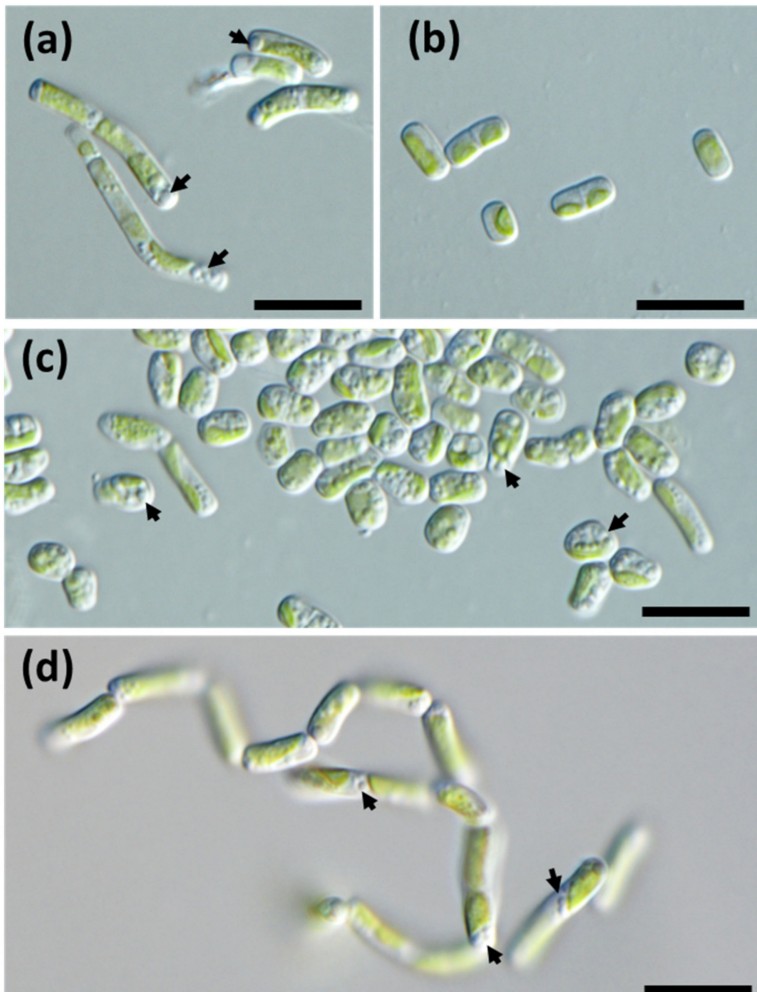

**Figure 2.** Morphological variability observed in *Diplosphaera elongata* sp. nov. by DIC microscopy: (**a**) elongated cells growing in liquid 3N-BBM + GC at 18 °C; (**b**) short-cylindrical cells in solid BBM at 18 °C; (**c**) pear-shaped cells in solid 3N-BBM + GC at 18 °C; (**d**) filament in liquid 3N-BBM + GC at 8 °C. Arrows indicate oil vesicles. Scalebars: 10 μm. Reference strain: ASUV 135.

*3.2. Molecular Phylogeny*

The resulting alignments included 75 taxa with 2898 characters for SSU + ITS (2155 + 743, respectively), and 54 taxa with 782 characteristics for *rbc*L. BI and ML phylogenetic hypotheses were topologically congruent on both trees.

In the concatenated SSU + ITS phylogenetic analysis, the strain *Diplosphaera* sp. ASUV135, designated here as *D. elongata*, was clustered with members of the *Diplosphaera* genus (Figure 3). In particular, *D. elongata* (ASUV135 strain) clustered with four sequences called *Diplosphaera* sp. 1 (OK636220, OK636221, OK636234 and OK636235) and another two uncultured sequences (KX221524 and MF481512) in a highly supported clade (Figure 3). The genus *Diplosphaera* proved to be monophyletic, as shown by a highly supported clade (ML/BI: 100/100). In this phylogeny, the genus *Diplosphaera* is divided into four clades that could be considered taxa in the species rank.

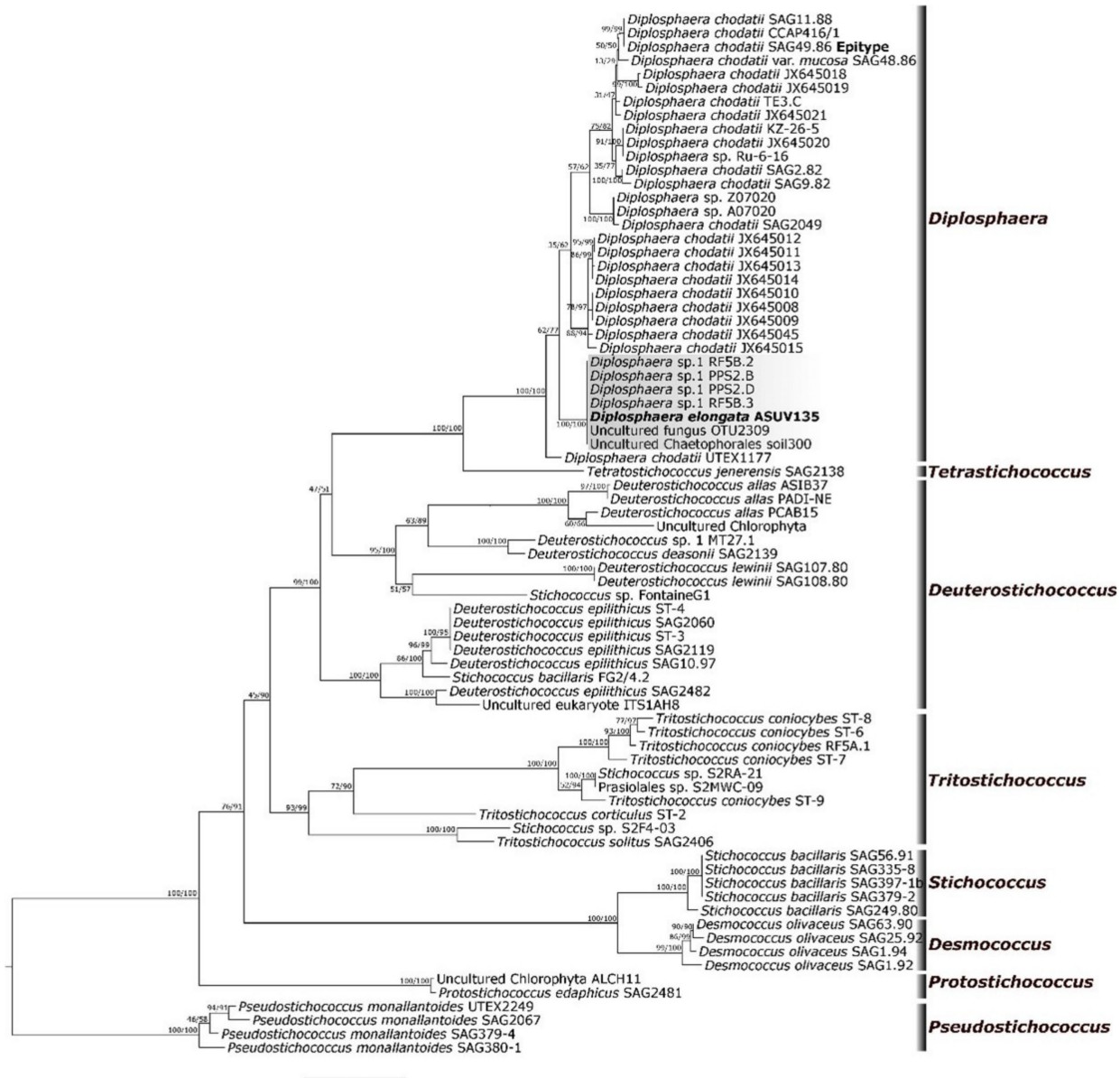

**Figure 3.** Phylogenetic tree based on the concatenated SSU and ITS dataset of 75 taxa of *Stichococcus*-like organisms. *Diplosphaera elongata* sp. nov. is highlighted in gray. Values at nodes indicate statistical ML bootstrap supports and Bayesian posterior probabilities (BS/PP). Scalebar shows the estimated number of substitutions per site.

The phylogeny of the *Stichococcus*-like organisms was also analyzed using the *rbc*L marker (Figure 4). As in the concatenated phylogeny, the genus *Diplosphaera* displays several clades that could be considered to be at the species rank. Again, the sequence of *D. elongata* (ASUV135) clustered with the genus *Diplosphaera*, specifically with *Diplosphaera* sp. L469 (JN573847), *Diplosphaera* sp. CG470 (JN573833) and *Diplosphaera* sp. AA53935 (JN573824) in a highly supported clade (Figure 4). *Diplosphaera chodatii* CCAP 416/1 clusters closely in a highly supported clade with *D. chodatii* SAG 49.86 (species epitype [5]) in both phylogenetic trees.

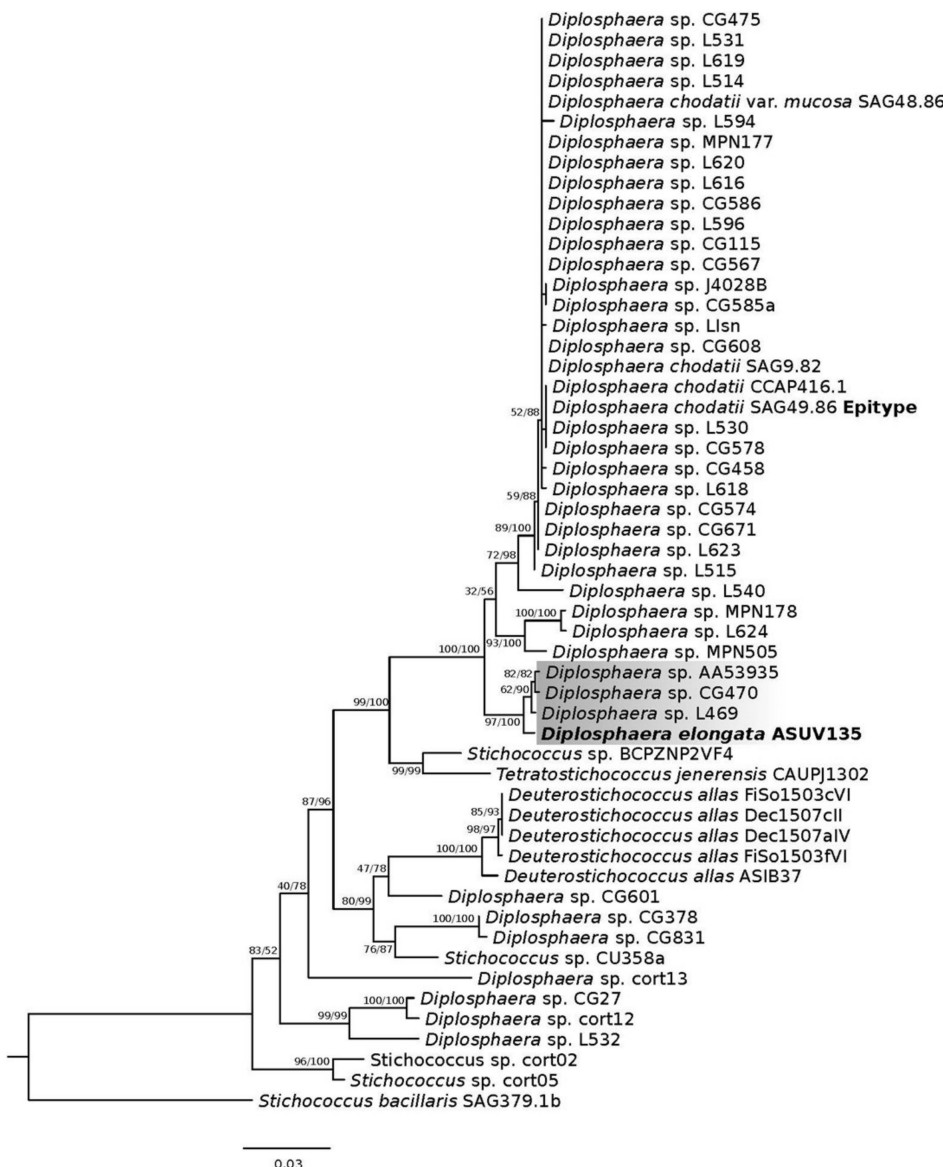

**Figure 4.** Molecular phylogeny based on *rbc*L sequences of 54 taxa of *Stichococcus*-like organisms. *Diplosphaera elongata* sp. nov. is highlighted in gray. Values at nodes indicate statistical ML bootstrap supports and Bayesian posterior probabilities (BS/PP). Scalebar shows the estimated number of substitutions per site.

### 3.3. ITS2 Secondary Structure

*Diplosphaera elongata* (ASUV135) differed from the genus type *D. chodatii* (SAG 49.86) by 39 nucleotides in the ITS2 primary sequence. A common overall organization of the ITS2 secondary structure could be identified in *D. elongata* (Supplementary Figure S1). The ITS2 secondary structures of *D. elongata* were compared to *D. chodatii* (authentic strain SAG 49.86 genus epitype) to check the occurrence of compensatory base changes (CBCs: nucleotide changes at both sides of the paired bases) and hemi-CBCs (changes at only one side of a nucleotide pair but still preserving pairing) according to Coleman [54]. One CBC was identified in helix II, and five hemi-CBCs were identified, one in helix I, one in helix II and three in helix IV (Supplementary Figure S1).

### 3.4. Ultrastructure and Morphology in Different Culture Media

The isolated microalga *Diplosphaera elongata* is characterized by rod-shaped cells (Figures 1, 2 and 5 and Supplementary Figure S2). The chloroplast is usually situated lon-

gitudinally close to the cell wall (Figures 1, 2 and 5 and Supplementary Figure S2). The cells may form filaments during reproduction via vegetative division (Figures 1a and 2d). The *D. elongata* (ASUV 135) cylindrical cells are highly variable in size (Figures 2 and 5). They range in length from 1.77 to 33.92 μm and in width from 1.67 to 4.27 μm. To assess cell morphology and phenotypic plasticity, the aspect ratio measurement was selected. In the different conditions analyzed, *D. elongata* (ASUV 135) showed noticeable differences in its AR, ranging from 1.4 to 1.9, and a maximum AR ranging from 2.3 to 3.8 (Figure 6 and Supplementary Table S2). When growing in 3N-BBM + GC, the mean AR was always over 1.7 (regardless of the temperature or type of culture, liquid or solid), reaching as high as 2.4 (18 °C, liquid), and the maximum AR ranged from 4.5 to 5.9 (Figure 6 and Supplementary Table S2). We must consider that AR will depend on the orientation of each individual cell in relation to the focal plane. Cells oriented longitudinally to the focal plane will display their true AR, whereas cells oriented perpendicularly will have an AR of around one. In the sixteen treatments used to test for phenotypic plasticity (Table 1), elongated cells were obtained as shown in Figures 1, 2, 5 and 6 and Supplementary Figure S2. Treatments with 3N-BBM + GC medium (except for solid 3N-BBM + GC at 18 °C) produced the most elongated cells compared with the other growth media (Figures 5 and 6 and Supplementary Figure S2). Although cells growing over solid 3N-BBM + GC at 18 °C were not particularly elongated, an anomalous thickening occurs, based on which they can be described as pear-shaped cells (Figures 2 and 5 and Supplementary Figure S2).

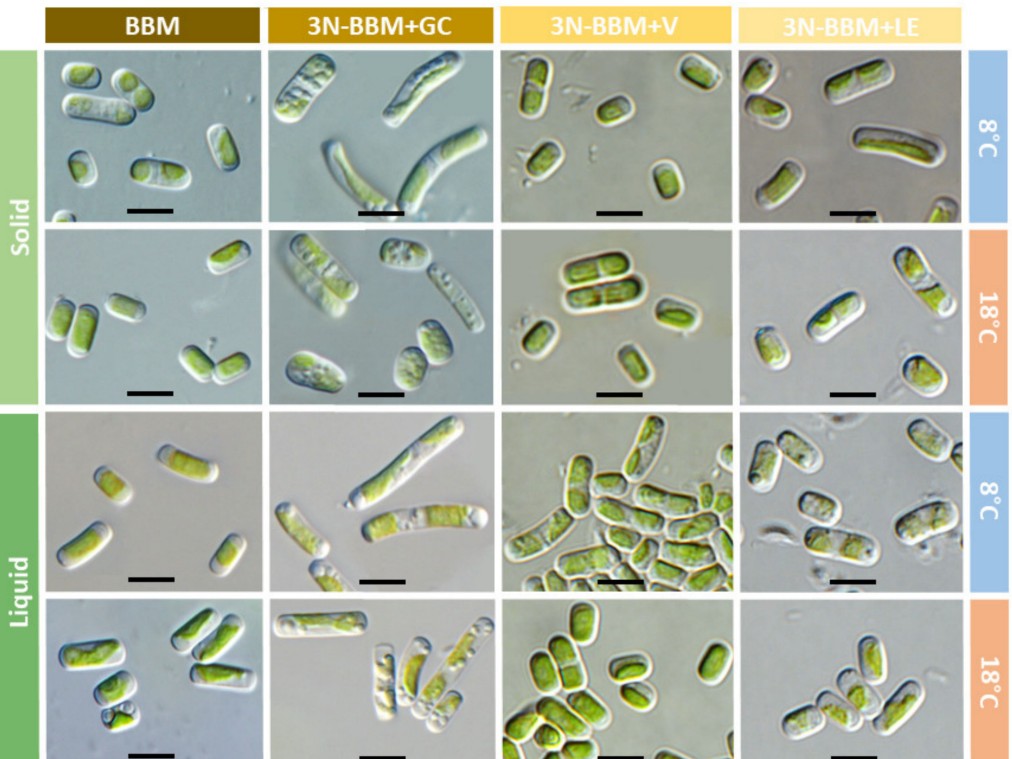

**Figure 5.** Morphology and phenotypic plasticity of *Diplosphaera elongata* sp. nov. after 21 days of growth under different conditions. Scalebars: 5 μm.

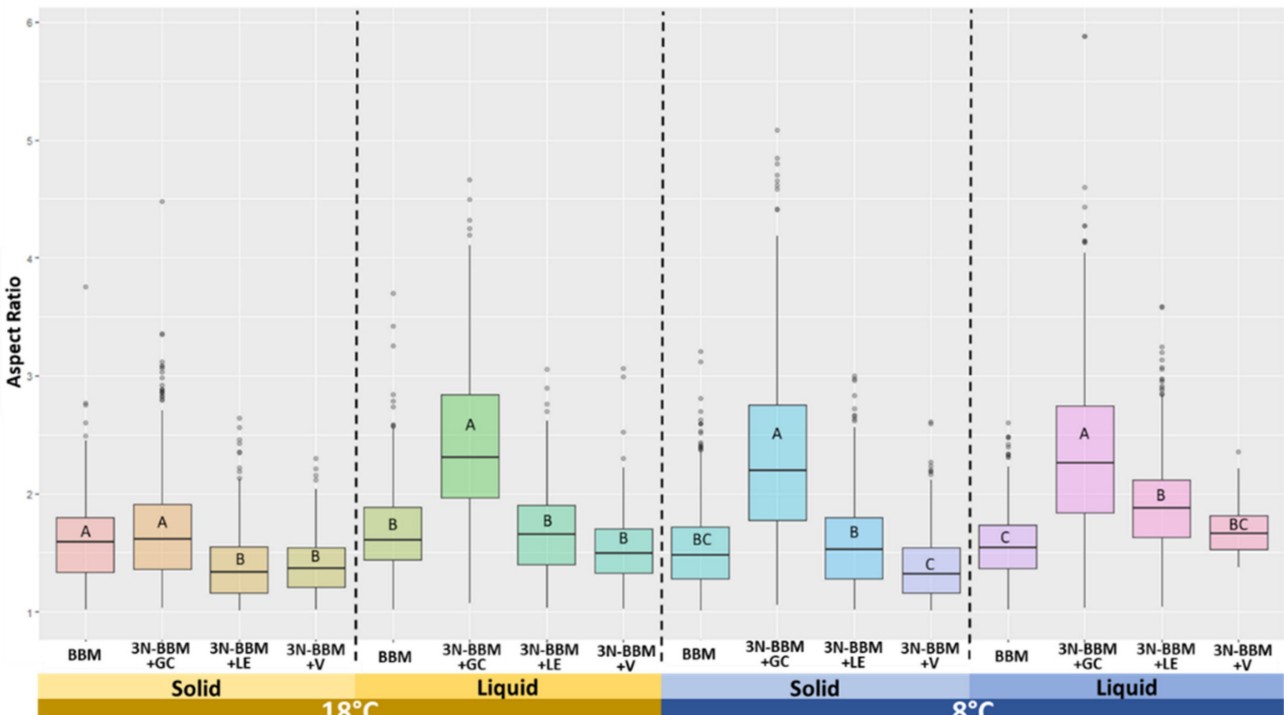

**Figure 6.** Boxplot of aspect ratio measurements to assess the morphology and phenotypic plasticity for each treatment. Combinations of temperature conditions and types of media are separated by dashed lines. Different letters indicate significant differences (Tukey test, $p < 0.05$) between culture media inside each combination of temperature and type of medium. The aspect ratio was calculated as Feret diameter (longest internal diameter)/longest internal diameter perpendicular to the Feret diameter.

CLSM was used to observe chloroplast morphology differences between BBM and 3N-BBM + GC media conditions at 8 °C and 18 °C (solid and liquid). The chloroplasts were located in the parietal position in all the conditions tested (Supplementary Figure S2). Of note was the presence of vesicles with high fluorescence in the rich medium at 18 °C in solid and liquid conditions (Supplementary Figure S2). In addition to the structure of the chloroplasts, the morphology of *D. elongata* cells can be observed by means of transmitted light images obtained using CLSM (Supplementary Figure S2). As with DIC images, every treatment with 3N-BBM + GC medium produced more elongated cells compared with the other growth media, except solid 3N-BBM + GC at 18 °C, which produced pear-shaped cells (Supplementary Figure S2).

The strain *D. chodatii* CCAP 416/1 was subjected to 3N-BBM + V, BBM and 3N-BBM + GC media at 8 °C (solid and liquid) to observe its cell morphology and possible phenotypic plasticity. Under all the conditions studied, this strain was observed showing a spherical or oval morphology. The appearance of cells with warts and brown cell walls could be observed when grown on the rich medium. Morphologically and phenotypically, *D. chodatii* CCAP 416/1 was different from *Dipolphaera elongata* ASUV135 under the different growth conditions used (Supplementary Figure S3).

### 3.5. Evaluation of Chlorophyll-Fluorescence Measurements

The form of the induction curves was similar in all the samples (Supplementary Figure S4). In Figure 7, the parameters obtained when fluorescence under actinic light was stable (average of the last four datapoints of each curve) are shown. $F_v/F_m$ provides an estimate of the maximum quantum efficiency of PSII photochemistry. On average, all the treatments showed an $F_v/F_m$ value of around 0.72–0.75, except for 3N-BBM + GC, grown at 18 °C, which had a lower value of 0.65 (Figure 7a). The effective quantum yield decreased in

all conditions when comparing 18° with 8 °C (Figure 7b), but the opposite was found for 3N-BBM + GC, where $\varphi_{PSII}$ was higher at 8° than at 18 °C. This was in line with the coefficient of photochemical quenching that estimates the fraction of open PSII centers [55] and showed the same pattern as $\varphi_{PSII}$ (Figure 7c).

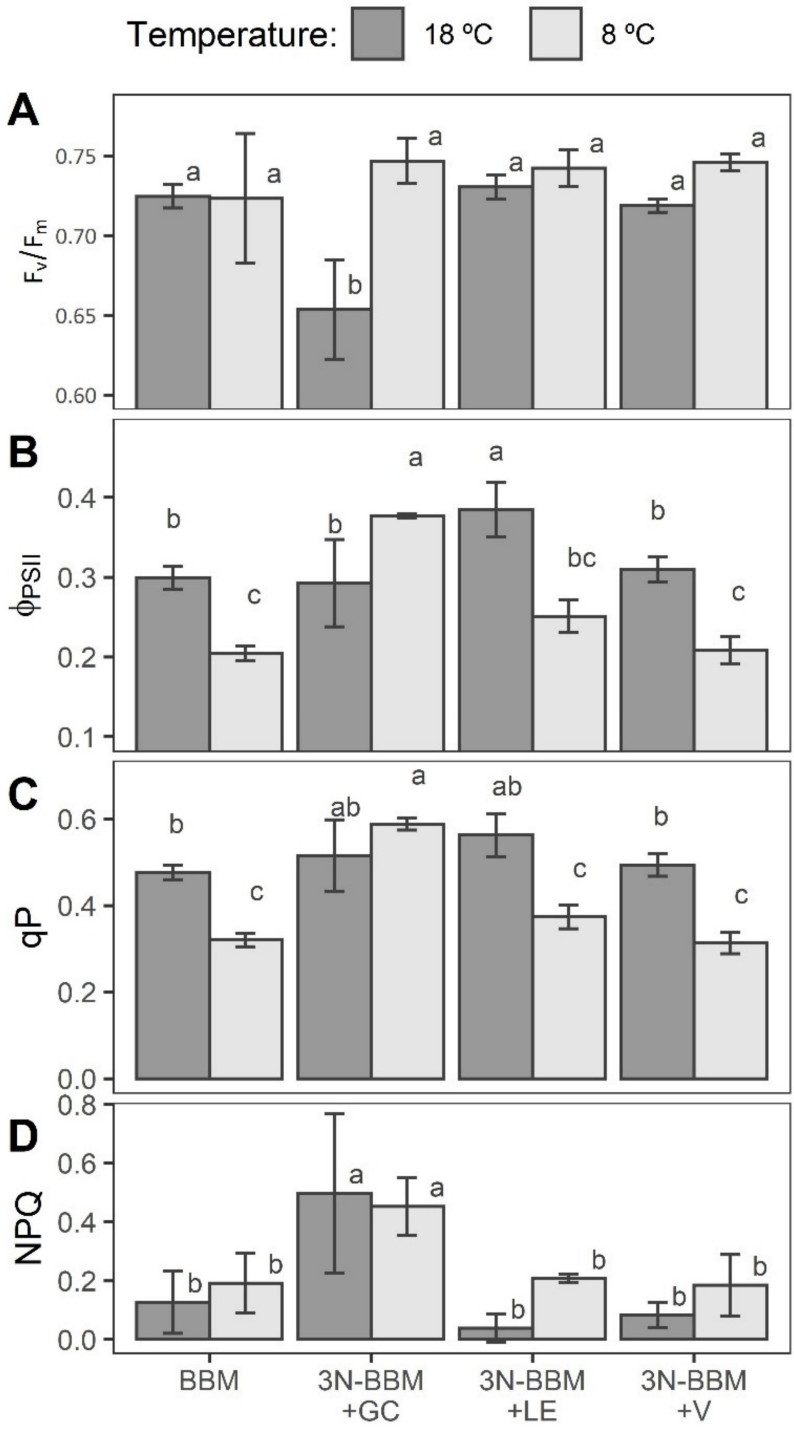

**Figure 7.** Chlorophyll *a* modulated fluorescence analysis of *Diplosphaera elongata* sp. nov. grown on different media and at different temperatures: (**A**) $F_v/F_m$, maximum quantum efficiency of PSII photochemistry; (**B**) $\varphi_{PSII}$, effective quantum yield; (**C**) qP, coefficient of photochemical quenching; and (**D**) NPQ, non-photochemical quenching parameter. Bars with the same letter are not statistically different (two-way ANOVA analysis, post hoc Tukey HSD test; $p < 0.05$; data represent mean ± standard deviation; $n = 4$).

No effect from the temperature treatments was indicated by the NPQ values (Figure 7d). However, the 3N-BBM + GC treatment presented higher values compared with the other conditions.

## 4. Discussion

Here, we formally describe *Diplosphaera elongata* as a new species of microalga isolated from lichen thalli. Although the strain used in this study (formerly, *Diplosphaera* sp. ASUV135) has been identified as a member of the lichen phycobiome, together with *Trebouxia* microalgae as the most frequent [22], we should not rule out the possibility that *D. elongata* could be detected as a primary microalga in some lichens. In addition, *D. elongata* has also been detected as a free-living microalga in New Zealand [53] and in a soil sample (GenBank code MF481512; direct submission without geographic information). Doering [56] reviewed the general ecological features of *D. chodatii* and determined that this microalga, in addition to being a symbiotic lichen partner, also grows on rocks at the interface between water and the aerial environment in the surroundings of rivers and lakes. Therefore, we can believe that this dual ability to grow as a both symbiotic and free-living organism is common to the species of the genus *Diplosphaera*, as is the case of *D. elongata*.

Molecular analysis (Figures 3 and 4 and Supplementary Figure S1) reveals that *D. elongata* is a new species of the genus *Diplosphaera*, despite belonging to a monotypic genus, *D. chodatti* [5]. Besides *D. elongata* and *D. chodatii*, there are clades in both trees with high robustness that could be considered candidate species. The *Stichococcus*-like organism tree inferred with the concatenation of the SSU and ITS nuclear genes was topologically congruent with those of previous studies [5,23,57] (Figure 3). The *rbc*L dataset could not be concatenated with SSU and ITS, so many of the new genera described by Pröschold and Darienko [5] could not be identified with this marker. Nevertheless, this marker allowed for the clear separation of the genus *Diplosphaera* from the remainder of the *Stichococcus*-like genera and the separation of *D. elongata* into a robust clade within the genus *Diplosphaera* (Figure 4). We wish to highlight the importance of the chloroplast marker *rbc*L as a discriminator between species or genera for the phylogenetic analysis of *Stichococcus*-like microalgae.

In addition to molecular analyses, this study carried out morphological, ultrastructural and physiological analyses to support the description of *D. elongata* as a new species. Up to three microscopy techniques were used to describe this species: DIC, CLSM microscopy and TEM (Figures 1, 2 and 5 and Supplementary Figure S2). TEM images have enabled us to understand the ultrastructure of the cells of *D. elongata* (Figure 1). With CLSM, we studied the shape and position of the chloroplast (Supplementary Figure S2). With DIC microscopy, we were able to study the size and length of the cells under each of the growth conditions used (Figures 2, 5 and 6). Medwed and coauthors [8] morphologically and ultrastructurally analyzed the strain CM01 using DIC and TEM images and found that the strain is spherical, similar to the other strains of *D. chodatii* [5]. *D. chodatii* CM01 shows a pyrenoid, numerous electron-dense vacuoles and a chloroplast with regularly arranged thylakoid membranes. On the other hand, in *D. elongate*, no pyrenoids were observed, nor were there any electron-dense vacuoles, and the thylakoids are not regularly organized (Table 2). Both species have a bi-layered cell wall and a parietal chloroplast in common. These two traits may need to be studied in the other *Diplosphaera* strains to determine if they are characteristic of the genus.

**Table 2.** Comparative table of the morphological, ultrastructural and phenotypical traits of *D. chodatii* and *D. elongata*.

| Trait | *Diplosphaera chodatii* | *Diplosphaera elongata* |
|---|---|---|
| Morphological | Individual cells are rather oval [5]. | Individual cells are rod-shaped. |
| Ultrastructural | Cells show a pyrenoid, numerous electron-dense vacuoles and a chloroplast with regularly arranged thylakoid membranes [8]. | Cells without pyrenoids or electron-dense vacuoles. In addition, the thylakoids are not regularly organized. |
| Phenotypical | Main factor: Temperature.<br>- At 18 °C, the cells show a spherical shape.<br>- At 8 °C, the cells show a slightly oval shape [5]. | Main factor: Medium composition.<br>- In sugar-rich media, the cells show an elongated rod shape.<br>- In non-sugar-rich media, the cells show a rod shape. |

The review by Sanders and Masumoto [20] highlights the controversy over the morphology of the *Diplosphaera* microalgae and the other *Stichococcus*-like organisms as morphologically plastic genera, which are often indistinguishable under the microscope. When we first observed the cylindrical cells of *D. elongata* ASUV 135 via DIC microscopy, we thought it was a *Stichococcus*-like organism, but phylogenetic analyses placed the strain in the genus *Diplosphaera* (Figures 3 and 4), despite it having a morphology not expected for this genus [5]. In the original description of this genus, Bialosuknia described the cells as spherical [18]. Subsequently, other morphological types have been reported for microalgae of the genus *Diplosphaera*, including micareoid, globose or cylindrical ones [14–16]. Based on this, it is worthwhile to point out that, in *Diplosphaera* microalgae, different cell morphologies have also been reported.

Pröschold and Darienko [5] outlined a phenotypic approach where *D. chodatii* cells are spherical, and, at colder temperatures (8 °C), the cells are more oval. In contrast, *D. elongata* cells have a cylindrical shape in all the growth conditions studied, both in cold (8 °C) and standard (18 °C) and solid or liquid conditions (Figure 5). They also change their appearance depending on the growth conditions (Figures 5 and 6). When this microalga is grown in a rich medium, cell length and size are increased in comparison with poor (BBM) or non-rich (3N-BBM + LE and 3N-BBM + V) media. The effects of temperature and medium density on the size of *D. elongata* are not significant. With these results, it can be assumed that *D. elongata* also shows phenotypic plasticity.

One of the most remarkable morphological changes occurs when *D. elongata* grows in rich medium (3N-BBM + GC), and its cylindrical cells become elongated and/or thickened (Figures 5 and 6). In addition, these cells contain several large oil droplets. Similar structures were observed by Proschold and Darienko [5] in some of the strains of *Stichococcus*-like genera, which are very prominent in *Deuterostichococcus* and *Pseudosticho-coccus*.

The modulated chlorophyll *a* fluorescence emissions were measured to characterize the photosynthetic characteristics of *D. elongata* under different growth conditions. In microalgae, as well as in plants, $F_v/F_m$ values are relatively constant in non-stressed cultures, and they usually decrease when organisms are subjected to stress [58–60]. The only treatment that showed a low value of $F_v/F_m$ was 3N-BBM + GC at 18 °C (Figure 7a). Together with the anomalous pear-shaped cells, this suggests that *D. elongata* is under abiotic stress while growing in these conditions. A decrease in $F_v/F_m$ can be due to a decrease in photochemistry or an increase in slow-relaxing non-photochemical quenching [50]. In the case of 3N-BBM + GC, the drop in $\varphi_{PSII}$ and qP when subjected to 8 °C suggests that the lower $F_v/F_m$ value may be explained by a decrease in photochemistry since NPQ levels did not show significant differences between the temperatures analyzed (Figure 7). The oil drops found when *D. elongata* was grown in the cited medium (Figure 5), independently of

the temperature, may be a storage strategy due to the excess of nutrients in the medium. Moreover, the higher level of energy dissipation, depicted by NPQ, compared with other media, supports the hypothesis of a lower need for photosynthetic efficiency under these conditions and, hence, the need to trigger a safety valve to protect the PSII reaction centers from damage due to excess irradiance [61].

*Diplosphaera chodatii* shows—in addition to a remarkable growth period on agar plates (68 days) compared with other symbiotic microalgae, e.g., *Asterochloris irregularis* (76 days) [62]—higher resistance to freeze/thaw cycles [17]. Moreover, *D. chodatii* shows high desiccation tolerance [8] and can survive up to two months in a dehydrated state, which, when symbiotizing, could help the lichens adapt to fluctuating water levels [63]. These features may be the key to explaining the dual capacity of this microalga to be both symbiotic and free-living. It cannot be ruled out that *D. elongata* may have similar characteristics due to the phylogenetic proximity of both species. It is likely that the high survival capacity of *Stichococcus*-like microalgae could explain their frequent presence in the phycobiomes of lichen thalli, even when they are not the primary photobionts [23].

In conclusion, our results evidence that *D. elongata* shows a phenotypic plasticity that is characteristic of the genus and highlight the importance of microscopic and physiological analyses under different growth conditions for its accurate description. Due to this plasticity, in *Stichococcus*-like organisms, the ultrastructural characterization must be supported by molecular techniques to achieve an integrated taxonomic approach. For *Stichococcus*-like genera, whose cells are morphologically similar, phenotypic behavior and photosynthetic measurements could be considered important characters for taxonomic circumscription.

**Supplementary Materials:** The following supporting information can be downloaded at https://www.mdpi.com/article/10.3390/d15020168/s1: Figure S1: *Diplosphaera elongata*. Secondary structure model of the internal transcribed spacer region (ITS2); Figure S2: Confocal reconstructions using CLSM and transmitted light imaging of *D. elongata*; Figure S3: The morphology of the strain *D. chodatii* CCAP 416/1 via DIC microscopy; Figure S4: Induction curves of the chlorophyll *a* fluorescence of *D. elongata*; Table S1: GenBank accession numbers for the species included in the phylogenetic analyses; Table S2: Aspect ratio (AR = width/height) expressed as the mean of each treatment.

**Author Contributions:** Conceptualization, S.C.; methodology, A.G., C.D.B. and S.C.; formal analysis, A.G., C.D.B. and S.C.; writing—original draft preparation, S.C.; writing—review and editing, A.G., C.D.B., E.B. and S.C.; supervision, E.B.; project administration, E.B.; funding acquisition, E.B. All authors have read and agreed to the published version of the manuscript.

**Funding:** This research was funded by PROMETEO 2021/005 (Prometeo Excellence Research Program, Generalitat Valenciana, Spain). S.C., received funding from a postdoctoral grant: Margarita Salas of the Ministerio de Universidades—Spain (Next generation EU, MS21-058). C.D.B., received funding from a postdoctoral grant from the Generalitat Valenciana and the European Social Fund (APOSTD19) and a María Zambrano grant from the Ministerio de Universidades (grant ZA21-046). A.G., received funding from a postdoctoral grant from the Generalitat Valenciana and the European Social Fund (APOSTD21).

**Institutional Review Board Statement:** Not applicable.

**Data Availability Statement:** See Section 3.1 Taxonomic Assessment.

**Acknowledgments:** Daniel Sheerin revised the English manuscript, and Alberto Martínez (Castellón) collaborated on the figure illustrations.

**Conflicts of Interest:** The authors declare no conflict of interest.

## Appendix A

Protocol for the preparation of the Bold Basal Medium with three-fold nitrogen and lichen extract (3N BBM + LE) [28,64,65].

**Table A1.** General purpose freshwater medium used for lichen axenic cultures. Preparation: To 940 mL of de-ionized or distilled water, add.

| ml | Stock Solution | g/400 mL dist. water | g/1000 mL dist. water |
|---|---|---|---|
| 10 | $NaNO_3$ | 10 | 25 |
| 10 | $CaCl_2 \cdot 2H2O$ | 1 | 2.5 |
| 10 | $MgSO_4 \cdot 7H_2O$ | 3 | 7.5 |
| 10 | $K_2HPO_4 \cdot 3 H_2O$ | 3 | 7.5 |
| 10 | $KH_2PO_4$ | 7 | 17.5 |
| 10 | NaCl | 1 | 2.5 |
| 6 | Microelement solution | | |
| 5 | Lichen extract (5%) | | |

**Table A2.** Preparation of the **microelement solution**: Add to 1000 mL of distilled water 0.75 g $Na_2$-EDTA and the minerals exactly in the following sequence.

| Ingredients | Quantity |
|---|---|
| $FeCl_3 \cdot 6H_2O$ | 97 mg |
| $MnCl_2 \cdot 4H_2O$ | 41 mg |
| $ZnCl_2 \cdot 6H_2O$ | 5 mg |
| $CoCl_2 \cdot 6H_2O$ | 2 mg |
| $Na_2MoO_4 \cdot 2H_2O$ | 4 mg |

Preparation of the **Lichen extract**: The lichen extract was prepared with 5 g of thalli (recently collected) that were rinsed for 1 min with sterile distilled water. Then, the lichen material was incubated with 100 mL of sterile Ringer's solution containing 0.05% Tween 20 surfactant (RST) on an orbital shaker (200 rpm) at room temperature. Later It was triturated in a blender and filter sterilized (including the use of 0.8-µm pore size polycarbonate pre-filters). For larger amounts of lichen extract preparation an additional centrifugation step ($6100 \times g$, Beckman Coulter JS-5.3 rotor) was included to ease pre-filtration and filter sterilization.

**Table A3.** Preparation of sterile **Ringer Solution** (1000 mL) [66].

| Ingredients | Quantity |
|---|---|
| NaCl | 2.25 g (dissolve in 200 mL) |
| KCl | 0.105 g (dissolve in 100 mL) |
| $CaCl_2$ | 0.06 g (dissolve in 100 mL) * |
| $NaHCO_3$ | 0.05 g (dissolve in 100 mL) |
| Distilled water | Up to 1000 mL |

* 0.079 g if $CaCl_2 \cdot 2H_2O$ (dihydrate form), Autoclave 20 min at 120 °C, Add 0.05% *v/v* Tween20 ® at the time of use.

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
