# Peer review of "Diplosphaera elongata sp. nova: Morphology and Phenotypic Plasticity of This New Microalga Isolated from Lichen Thalli"

_diversity, doi:10.3390/d15020168_

Round 1
Reviewer 1 Report
Please see all comments and suggestions in the attached file.

Author Response
Reviewer 1
Comments on diversity-2127995 entitled “Diplosphaera elongata sp. nova: Morphology and phenotypic plasticity of this new microalga isolated from lichen thalli”
This manuscript presents description of new species, Diplosphaera elongate (Trebouxiaceae), isolated from a lichen thallus. A comprehensive analysis of this taxon has been carried out using an integrative taxonomic approach, including phylogenetic and ITS2 secondary structure analysis; study of morphology, ultrastructure and phenotypic plasticity; exploring of modulated chlorophyll a fluorescence emissions. The results are interesting and well-presented, and allow to considering the described taxon as a new species. I have some suggestions for improving this manuscript.
Firstly, Diplosphaera sphaerica (Tschermak-Woess) Karbovska & Kostikov is not synonym of Diplosphaera chodatii Bialosuknia. According to Darienko & Pröschold (2019) (line 48-49), it is a new species of another genus – Jaagichlorella sphaerica (Tschermak-Woess) Darienko & Pröschold.
See: Darienko, T. & Pröschold, T. The genus Jaagichlorella Reisigl (Trebouxiophyceae, Chlorophyta) and its close relatives: an evolutionary puzzle. Phytotaxa. 2019. 388(1): 47–68. https://www.algaebase.org/search/species/detail/?species_id=171339
Thanks for the correction. Change made to the manuscript. See lines 41-48.
Secondly, the data from section “3.2. ITS2 secondary structure” contradict with the
Supplementary Figure S1. The authors point out that “Diplosphaera elongata (ASUV135)
differed from the genus type D. chodatii (SAG 49.86) by 39 nucleotides in the ITS2 primary
sequence” (lines 367-368). But the Supplementary Figure S1 shows only 22 substitutions.
Further “Four CBCs were identified, three in helix I and one in helix IV; and three hemi-CBC
were identified, one in helix II and two in helix IV (Supplementary Figure S1)” (lines 373-375).
However only 2 CBC are presented in helix I. One marked CBC is not CBC, because this is not
nucleotide changes but deletions of nucleotides at both sides of paired bases. One hCBC is not
marked in a green box in helix IV in the Supplementary Figure S1. The text in Section 3.2 and
the Supplementary Figure S1 should be corrected.
Thanks. We have corrected the text in Section 3.2 and Supplementary Figure S1.
Other minor revisions:
1) Lines 86, 192, 194, 210, 215, 216, 279, 329, 345, 358, 498, 505: “rbcL” should be
corrected to “rbcL”. Done.
2) Lines 103, 191: “ITSrDNA” should be corrected to “ITS rDNA”. Done.
3) Line 111-112: “50 μmol/ m−2s−1” should be corrected to “50 μmol/m−2s−1”. Done.
4) Line 136: For a light microscope and a camera, you should give the name of the
manufacturer, city and country. Done.
5) Line 159-160: For a laser scanning confocal microscope, you should give the name of
the manufacturer, city and country. Done.
6) Line 171: “OsO4” should be corrected to “OsO4”. Done.
7) Line 190: “SSUrDNA” should be corrected to “SSU rDNA”. Done.
8) Line 282: “Diplosphaera chodatii cells” should be corrected to “D. chodatii cells”. Done.
9) Line 590: “(ITS)2” should be corrected to “(ITS2)”. Done.
10) Line 591: “D. elongate” should be corrected to “D. elongata”. Done.
11) Line 592: “chlorophyll a” should be corrected to “chlorophyll a”. Done.
12) Line 618: “(2018)” should be deleted. Done.
13) Line 690: “(2021)” should be deleted. Done.
14) Line 691: “Watanabea lichenicola” should be corrected to “Watanabea lichenicola”. Done.
15) Line 781: “(2008)” should be deleted. Done.
16) The numbering in References should be aligned. Done.
17) Many references are missing DOI. It should be added. Done.
Reviewer 2 Report
Overall comments
The authors provide multiple lines of evidence for the separation from the new taxon Diplosphaera elongata from the other members of the genus. The authors use technically sound methods to clarify the taxonomic status of the unknown strains. The presented results are sufficient to warrant a new taxonomic status for the studied taxa. However, for the benefit of broader audience, it would be nice to see additional explanations as to why the phenotypic plasticity and photosynthetic characters have to be measured in a taxonomic description. The reader would also find it helpful to see a figure and/or a table summarizing the comparisons the new species with the D. chodatii that the authors have provided throughout the manuscript
Specific comments
- L.18-19: the sentence about the phenotypic plasticity is not clear. Also it is not immediately clear why we would be interested in phenotypic plasticity in describing a new species.
- L.33-36: The broader audience might not be familiar with the "green puzzle" mentioned here. More context for the taxonomic problems in this group would be great. Also, the use of "this study" throughout the manuscript is a bit unclear whether it refers to "the cited study" or the current manuscript.
- L.48-50: Why were these names synnonymized? The breif explanation will provide a good context for estabilishing a new taxon.
- L. 61-62: It would be nice to also conclude why the phenotypic plasticity is important for taxonomic circumscription in microalgae.
- L. 84: The origin of the studied strain should be mentioned before or at the objective of the study.
- L. 88-89: "the kinetics of the modulated chlorophyll a flurescence emission" is probably not the right word, as the measurements here don't really show the kinetics (changes over time) per se). Also "to understand their behavior" is a bit too broad.
- L. 115-122: The abbreviations for the components in the medium were not explained.
- L. 149-150: What were these cultures used for the comparison with the other species?
- L. 260: Please indicate the version of R used in the study.
- L. 264-266: Please check with the taxonomic name convention of the journal, e.g. The author names should not be italicized.
- L. 327: Figure 3 showed that most of GenBank's Diplosphaera rbcL seqeunces were not identified to the species, while the tree from nuclear data placed the new species within the D. chodatii group (D. chodatiiUTEX1177 was sister to everything else). This discrepancy needs some clarification.
- L. 394: The explanation about the minimum AR being 1 is more belonged to the Method section.
- L. 464-467: These lines belong to the method or the discussion.
- L. 498-500: This explanation for not concatenating the dataset should go earlier in the method seciton.
- L. 552-553: This sentence might be overly complex. How about: "The modulated Chlorophyll a fluorescence emissions were measure to characterize the photosynthetic characters of D. elongata under different growth conditions."
Table and Figure
- Table 1: would it be better to organize it in a way that each type of conditions is directly comparable? The controlled factors like number of days and light periods can be explained in the table caption instead (see Also the abbreviations (N, BBM, V etc.) have to be explained in the table as well. For example >
| Treatment | Standard condition | Poor condition | Rich condition | Lichen condition |
|---|---|---|---|---|
| 18°C | 3N-BBM+V | |||
| 8°C | 3N-BBM+V | |||
| --------- | --------- | --------- | --------- | --------- |
- Figure 3: Why should we include "uncultured fungus OTU2309" in the analysis? Is this the annotation from NCBI database, or our own? Also the number on the scale bar was cut off in the PDF version.
- Figure 5: The names of the conditions here are not consistent with the naming in Table 1. If we decide to go with the names of the media, it might not be necessary to include table 1.
- Figure 6: The axis labels were a bit small. A short description what asepct ratio is within the caption would also be nice.
- Figure 7: Explanations of the letters (a,b,c) and the error bars should be explained in the caption as well. Also NPQ of the 3N-BBM+LM seems like it could be significantly different from the other treatments.
Author Response
Reviewer 2
The authors provide multiple lines of evidence for the separation from the new taxon Diplosphaera elongata from the other members of the genus. The authors use technically sound methods to clarify the taxonomic status of the unknown strains. The presented results are sufficient to warrant a new taxonomic status for the studied taxa. However, for the benefit of broader audience, it would be nice to see additional explanations as to why the phenotypic plasticity and photosynthetic characters have to be measured in a taxonomic description. The reader would also find it helpful to see a figure and/or a table summarizing the comparisons the new species with the D. chodatii that the authors have provided throughout the manuscript
Thanks. In the manuscript we have highlighted why phenotypic plasticity and photosynthetic characters should be considered in a taxonomic description. In addition, a comparative table between D. chodatii and D. elongata has been provided.
Specific comments
L.18-19: the sentence about the phenotypic plasticity is not clear. Also it is not immediately clear why we would be interested in phenotypic plasticity in describing a new species.
Done. See lines 18-20.
L.33-36: The broader audience might not be familiar with the "green puzzle" mentioned here. More context for the taxonomic problems in this group would be great. Also, the use of "this study" throughout the manuscript is a bit unclear whether it refers to "the cited study" or the current manuscript. Done. See lines 31-35.
L.48-50: Why were these names synnonymized? The breif explanation will provide a good context for estabilishing a new taxon. Done. See lines 44-48.
- 61-62: It would be nice to also conclude why the phenotypic plasticity is important for taxonomic circumscription in microalgae. Done. See lines 59-62.
- 84: The origin of the studied strain should be mentioned before or at the objective of the study. Done. See line 85.
- 88-89: "the kinetics of the modulated chlorophyll a flurescence emission" is probably not the right word, as the measurements here don't really show the kinetics (changes over time) per se). Also "to understand their behavior" is a bit too broad. Done. See lines 88-90.
- 115-122: The abbreviations for the components in the medium were not explained. Done. See lines 117-120.
- 149-150: What were these cultures used for the comparison with the other species?
At low temperatures (8°C) the change of morphology (phenotypic plasticity) of the tested D. chodatii strains by Pröschold and Darienko was demonstrated [5]. For this reason, 3N-BBM+V, BBM and 3N-BBM+GC media at 8°C in solid and liquid media were selected in this study for a morphological comparison of D. chodatii CCAP 416/1 with D. elongata ASUV135. This would allow us to clearly demonstrate that D. elongata and D. chodatii have differences in their phenotypic plasticity.
- 260: Please indicate the version of R used in the study. Done. See line 270.
- 264-266: Please check with the taxonomic name convention of the journal, e.g. The author names should not be italicized. Done.
- 327: Figure 3 showed that most of GenBank's Diplosphaera rbcL seqeunces were not identified to the species, while the tree from nuclear data placed the new species within the D. chodatii group (D. chodatiiUTEX1177 was sister to everything else). This discrepancy needs some clarification.
As can be seen in Figure 3 and Supplementary Table 1, many of the Rbcl sequences of Stichococcus-like organisms are not well assigned in the GenBank (GB) to any current taxon. Most are named as Diplosphaera sp. or Stichococcus sp., and while some sequences undoubtedly belong to the genus Diplosphaera, many others must belong to other Stichococcus-like genera. Furthermore, very few strains have sequences deposited from both markers (ITS and rcbL) in the GB, making it impossible to concatenate both datasets. This concatenation would possibly allow us to classify rcbl sequences without GB information into one of the other Stichococcus-like genera. Therefore, the comparison between the two phylogenies (Fig 3 and 4) is limited. Nevertheless, both phylogenies allow us to clearly delimit the sequences belonging to the genus Diplosphaera. The other genera in the rcbl tree cannot be identified with the information available in the GB. As for D. chodatii UTEX1177, only ITS information is available (see supplementary Table 1). There is no rbcL information on this strain and for this reason it is not included in the tree in Figure 4. I personally believe that D. chodatii UTEX1177 is another new species of Diplosphaera, different from D. chodatii and D. elongata, but we do not have enough information to propose this in this publication.
- 394: The explanation about the minimum AR being 1 is more belonged to the Method section. Done. See lines 152-154.
- 464-467: These lines belong to the method or the discussion. Done. We added this information to the Material and methods. Lines 257-261.
- 498-500: This explanation for not concatenating the dataset should go earlier in the method seciton. Done. See lines 212-217, 490 and 492.
- 552-553: This sentence might be overly complex. How about: "The modulated Chlorophyll a fluorescence emissions were measure to characterize the photosynthetic characters of D. elongata under different growth conditions." Done, thanks. See lines 549-550.
Table and Figure
Table 1: would it be better to organize it in a way that each type of conditions is directly comparable? The controlled factors like number of days and light periods can be explained in the table caption instead (see Also the abbreviations (N, BBM, V etc.) have to be explained in the table as well. For example >
Treatment Standard condition Poor condition Rich condition Lichen condition
18°C 3N-BBM+V
8°C 3N-BBM+V
--------- --------- --------- --------- ---------
Done. Table 1 reorganised.
Figure 3: Why should we include "uncultured fungus OTU2309" in the analysis? Is this the annotation from NCBI database, or our own? Also the number on the scale bar was cut off in the PDF version.
This sequence is not ours. This is the annotation given by the submitter when the sequence was uploaded to the NCBI database. The submitter incorrectly identified as a fungus a sequence that is clearly an alga. After publication of this article, with the formal description of D. elongata, we will contact the GenBank to assign this new taxon to the sequence uncultured fungus OTU2309 and the rest of the sequences that are not well classified. I will contact the editor to point out that the number in the scale bar was cut off.
Figure 5: The names of the conditions here are not consistent with the naming in Table 1. If we decide to go with the names of the media, it might not be necessary to include table 1.
Done, thanks. When I modified Table 1 based on your previous suggestion I made the corrections to make this table consistent with Figure 5. See Table 1.
Figure 6: The axis labels were a bit small. A short description what asepct ratio is within the caption would also be nice. Done. See Figure 6 and Caption 6.
Figure 7: Explanations of the letters (a,b,c) and the error bars should be explained in the caption as well. Also NPQ of the 3N-BBM+LM seems like it could be significantly different from the other treatments.
We added the information in Caption 7. Regarding NPQ that could be significant, the data was analysed by a two-way ANOVA in which the effects of the medium, temperature, and the interaction between the medium and temperature were tested. In all the tested parameters, the interaction showed a p-value < 0.05 and, hence, a Tuckey post-hoc test was performed. However, in the case of the parameter NPQ, the interaction between the medium and temperature showed a p-value > 0.05 and only the effect of the medium proved to be significantly different. This is probably because the sample size is small.